# Deep learning for obstructive sleep apnea diagnosis based on single channel oximetry

Jeremy Levy[1,2], Daniel Álvarez [3,4,5], Félix Del Campo[3,4,5] & Joachim A. Behar [2] ✉

Obstructive sleep apnea (OSA) is a serious medical condition with a high prevalence, although diagnosis remains a challenge. Existing home sleep tests may provide acceptable diagnosis performance but have shown several limitations. In this retrospective study, we used 12,923 polysomnography recordings from six independent databases to develop and evaluate a deep learning model, called OxiNet, for the estimation of the apnea-hypopnea index from the oximetry signal. We evaluated OxiNet performance across ethnicity, age, sex, and comorbidity. OxiNet missed 0.2% of all test set moderate-to-severe OSA patients against 21% for the best benchmark.

Obstructive sleep apnea (OSA) is a highly prevalent condition. Benjafield et al.[1] estimated moderate-to-severe OSA affects 425 million (95% CI 399–450) adults aged 30–69 years in a review including 17 studies. OSA is characterized by recurrent episodes of upper airway partial or complete upper airway obstruction associated with recurrent oxyhemoglobin desaturations (intermittent hypoxia) and arousals (sleep fragmentation). It is caused by upper airway collapse during sleep and is characterized by frequent awakenings caused by apnea and/or hypopnea. Several studies have shown that if untreated, OSA increases the risk of cardiovascular diseases, stroke, death, cancer, and other diseases[2], which all bear high clinical and economical costs. The most common presenting symptom of OSA is excessive daytime sleepiness[2], leading to accidents and less effective work. Currently, full-night polysomnography (PSG), is considered the gold standard for confirming the clinical suspicion of OSA, assessing its severity, and guiding therapeutic choices. It is a multi-channel monitoring technique that analyzes the electrophysiological and cardio-respiratory patterns of sleep.

In the past year, home sleep apnea tests (HSAT) have emerged as an alternative to in-lab PSG. Although HSAT became standard practice in some countries, a recent meta-analysis of 20 papers revealed a misdiagnosis rate of 39%[3] in HSAT thus highlighting a gap in the performance of these alternative cost-efficient solutions and the more expensive gold standard in-lab PSG. Several factors have motivated the development of portable diagnostic technology such as those based

on single channel oximetry analysis[4–8]: the growing awareness of the high prevalence of OSA, the high proportion of undiagnosed individuals[9] but also low time, cost-effectiveness and low availability of PSG and the limited performance of existing HSAT. However, most previous studies[4,5] focused on developing OSA screening tests with a binary classification task (OSA, non-OSA). In some cases[6,8,10], a multiclass classification task was considered to take into account all degrees of severity of OSA, but failed to estimate the apnea–hypopnea index (AHI).

In this work, we develop and evaluate the robustness of a single-channel oximetry-based OSA diagnosis algorithm based on deep learning (DL), called OxiNet, in multiple distribution shifts. OxiNet is benchmarked against two state-of-the-art classical machine learning (ML) approaches in the field of oximetry analysis for OSA diagnosis. OxiNet, significantly outperformed benchmark algorithms on all external test datasets while missing 0.2% of all test sets moderate-to-severe patients against 21% for the best benchmark. The main drops in performance on external test sets were due to a distribution shift in ethnicity for Black participants and African American participants, and comorbidity for individuals with concomitant chronic obstructive pulmonary disease.

## Results

We used 12,923 PSG recordings, totaling 115,866 h of continuous data, from 6 independent databases to develop and evaluate OxiNet for the

[1]The Andrew and Erna Viterbi Faculty of Electrical & Computer Engineering, Technion-IIT, Haifa, Israel. [2]Faculty of Biomedical Engineering, Technion, Israel Institute of Technology, Haifa, Israel. [3]Río Hortega University Hospital Valladolid, Valladolid, Spain. [4]Biomedical Engineering Group, University of Valladolid, Valladolid, Spain. [5]Centro de Investigación Biomédica en Red en Bioingeniería, Biomateriales y Nanomedicina (CIBER-BBN), Valladolid, Spain. ✉e-mail: jbehar@technion.ac.il

regression task of AHI estimation from the oximetry signal. The AHI was defined as the average number of all apneas and hypopneas, according to the recommended rule, per hour of sleep following the American Academy of Sleep Medicine (AASM) 2012 rules[11], and ICSD-3 guidelines[12]. Recordings with technical faults, with $\widehat{TST} < 4$ (i.e. less than 4 h of sleep) and patients under 18 years old were excluded. We assess OxiNet performance across ethnicity, age, sex, and comorbidity. We benchmark OxiNet against an ML model using the oxygen desaturation index (ODI) as input and an ML model using digital oximetry bio-markers (OBM) as input.

Table 1 presents the size of each dataset before and after applying the exclusion criteria. SHHS1 and SHHS2 consist of the same cohort (longitudinal study) and thus in order to avoid information leakage, the recordings of the SHHS1 training set were discarded in SHHS2 yielding 621 recordings listed under the "original number of record-ings" in SHHS2.

The model with the lowest loss in the validation set was saved for inference on the test sets. Table S1 presents the results of OxiNet trained on 90% of SHHS1. OxiNet achieved ICC = 0.96, $F_{1,M}$ = 0.84 for the SHHS1 test set and ICC = 0.95, $F_{1,M}$ = 0.83 for SHHS2. Performance was impaired but acceptable for UHV (ICC = 0.95, $F_{1,M}$ = 0.77), CFS (ICC = 0.92, $F_{1,M}$ = 0.78), MROS (ICC = 0.94, $F_{1,M}$ = 0.80) and MESA (ICC = 0.94, $F_{1,M}$ = 0.75). In all external databases, OxiNet performance was consistently and significantly better than the ODI and OBM benchmark models (Table S2). P-values of Wilcoxon rank-sum tests between OxiNet and OBM were below 0.05 for all databases. There was no statistical difference in performance between men and women. Table 2 presents the ICC and $F_{1,M}$ for all the databases and the different trained models. Figure 1 compares the estimated AHI against the actual AHI, for all the external databases.

## Table 1 | Size of each database before and after applying the exclusion criteria

| Database | Original number of recordings | Technical fault | $\widehat{TST} < 4$ | Age < 18 years | Recordings after exclusion criteria |
|---|---|---|---|---|---|
| SHHS1 | 5778 | 59 (1%) | 99 (2%) | 0 (0%) | 5620 (97%) |
| SHHS2 | 621 | 0 (0%) | 16 (3%) | 0 (0%) | 605 (97%) |
| UHV | 369 | 0 (0%) | 12 (3%) | 0 (0%) | 357 (97%) |
| CFS | 728 | 13 (2%) | 71 (10%) | 58 (8%) | 586 (80%) |
| MROS | 3937 | 51 (1%) | 133 (3%) | 0 (0%) | 3753 (95%) |
| MESA | 2056 | 0 (0%) | 54 (3%) | 0 (0%) | 2002 (97%) |

74% of SHHS2 were removed, because the patients from SHHS1-train were excluded to avoid any leakage information.

## Discussion

Efforts focused on the analysis of respiratory pathologies based on oximetry time series have received considerable attention in the last few years[13]. Numerous studies have proposed oximetry biomarkers that describe patterns present in the oximetry signal, such as approx-imate entropy[14], detrended fluctuation analysis[15] or desaturations-based biomarkers[16]. Hang et al.[4] proposed a support vector machine model which takes handcrafted features as input. They trained their model on a total of 699 patients with suspected OSA and reported a sensitivity of 0.89 for severe OSA and 0.87 for moderate-to-severe OSA. Behar et al.[5] developed OxyDOSA, which is a linear regression model trained on oximetry biomarkers and three clinical features. They trained the model on a clinical PSG database of 887 individuals from a representative São Paulo (Brazil) population sample. They performed a binary classification of non-OSA versus OSA and obtained an AUROC of 0.94 ± 0.02, the sensitivity of 0.87 ± 0.04, 0.99, and 1.00 for the test set, moderate, and severe OSA respectively. Using the SHHS database, Deviaene et al.[6] trained a random forest model based on 139 $SpO_2$ features and 4 clinical features, with the aim of classifying 1-min seg-ments as having or not a desaturation within them or not. They obtained an average sensitivity of 0.64 on the SHHS1 test set for the binary classification task and 67.0% for the multi-class classification task. When training an ensemble learning model based on features extracted from the oximetry signal, Gutierrez et al.[8] achieved a kappa score between 0.45 and 0.66 when considering the four OSA severity categories, working on a database composed of 8,762 recordings. Previous studies mostly feature engineering-based models involving oximetry biomarkers and some clinical variables[5,6,8,13]. Mostafa et al.[17] proposed a DL approach for the detection of sleep apnea using oxi-metry, but built their model on data from 33 patients only, achieving an accuracy of 0.97 and a sensitivity of 0.78. Despite the large number of studies focused on OSA diagnosis from oximetry data, they suffer cri-tical limitations that have led to inconclusive beliefs regarding the viability of applying oximetry for OSA diagnosis. These limitations included the limited performance of the ML models developed, the experimental setting defining the challenge as a multi-class classifica-tion task thus preventing the assessment of such models for diagnostic purposes. In our contributions, we used a total of 12,923 PSG record-ings, totaling 115,866 h of continuous data, from five independent databases to develop a robust DL model, denoted OxiNet, for the estimation of the AHI and to address research gaps in assessing the robustness of such an algorithm across ethnicity, age, sex, comorbid-ity, and medical guidelines. This research makes two main contribu-tions. The first is the creation of OxiNet, a robust DL model for the estimation of the AHI from oximetry time series. The second is

## Table 2 | Results on the test set of the external databases

| | ODI | | OBM | | OxiNet | |
|---|---|---|---|---|---|---|
| | ICC | $F_{1,M}$ | ICC | $F_{1,M}$ | ICC | $F_{1,M}$ |
| SHHS1 | 0.89 | 0.69 | 0.93 | 0.74 | 0.96 | 0.84 |
| | (0.89–0.93) | (0.68–0.72) | (0.90–0.95) | (0.70–0.78) | (0.95–0.97) | (0.82–0.86) |
| SHHS2 | 0.89 | 0.69 | 0.93 | 0.74 | 0.95 | 0.83 |
| | (0.87–0.91) | (0.65–0.72) | (0.92–0.93) | (0.74–0.75) | (0.94–0.98) | (0.83–0.85) |
| UHV | 0.75 | 0.61 | 0.86 | 0.67 | 0.92 | 0.77 |
| | (0.74–0.78) | (0.58–0.62) | (0.88–0.92) | (0.60–0.74) | (0.91–0.96) | (0.77–0.79) |
| CFS | 0.70 | 0.56 | 0.75 | 0.60 | 0.92 | 0.78 |
| | (0.66–0.78) | (0.50–0.70) | (0.68–0.80) | (0.51–0.69) | (0.90–0.96) | (0.74–0.82) |
| MROS | 0.70 | 0.52 | 0.81 | 0.65 | 0.94 | 0.80 |
| | (0.68–0.72) | (0.50–0.58) | (0.79–0.83) | (0.62–0.68) | (0.95–0.99) | (0.78–0.84) |
| MESA | 0.75 | 0.60 | 0.75 | 0.65 | 0.94 | 0.75 |
| | (0.71–0.77) | (0.54–0.64) | (0.70–0.80) | (0.62–0.68) | (0.92–0.94) | (0.72–0.76) |

ODI, OBM, and OxiNet are trained on 90% of SHHS1. Confidence intervals are computed via bootstrapping on each test set separately.

# OxiNet

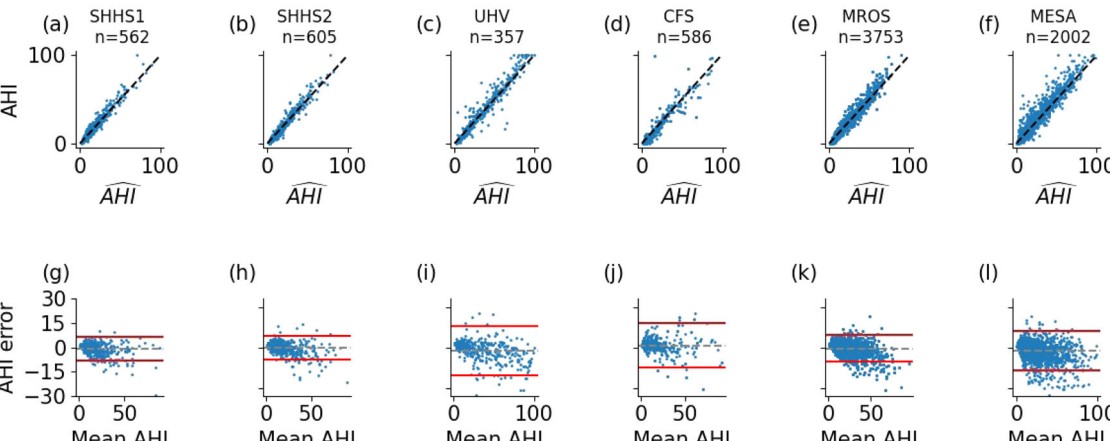

**Fig. 1 | Correlation and Bland–Altman plots.** Top (**a–f**): scatter plot of the computed and annotated AHI for all external databases. The dotted line represents the equation $y = x$. Bottom (**g–l**): Bland–Altman plot between the estimated and the annotated AHI. The error lines are positioned at ± 1.96 the standard deviation. From left to right: SHHS1, SHHS2, UHV, CFS, MROS, and MESA. $R^2$ statistics are summarized in Table 2.

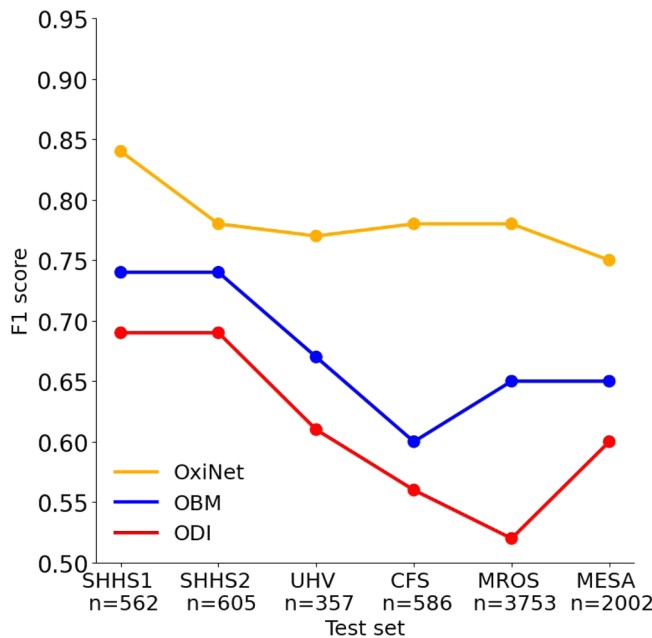

**Fig. 2 | Models' performance.** $F_1$ score for each model as a function of the test set.

the performance assessment of OxiNet across different databases and distribution shifts.

The first main contribution of the research is the creation of OxiNet, a robust DL algorithm for the estimation of the AHI. OxiNet is shown to significantly outperform benchmark feature engineering based algorithms (ODI and OBM) on all test databases (Fig. 2 and Table 2). The baseline performance was determined by training a model taking the ODI as the sole oximetry feature. Indeed, ODI has been historically the most studied and used single oximetry based feature for OSA screening. The performance of the ODI based model on the SHHS1 test set was poor (ICC = 0.89, $F_1$ = 0.69, Fig. 2) and would have led to 55 missed moderate to severe OSA diagnosis (21% of all moderate and severe). This demonstrates that using the ODI as the sole oximetry feature, i.e., only considering the average number of desaturations per hour, is not sufficient to enable robust AHI estimation

and thus OSA diagnosis. When combining the set of 178 OBMs within a CatBoost model the performance increased significantly on the SHSH1 test set (ICC = 0.93, $F_1$ = 0.74). Yet some important miss-classification errors remained with 971 missed moderate to severe OSA patients across all the databases (21% of all moderate and severe, Table 2). Our algorithm OxiNet performed significantly better on SHHS1 (ICC = 0.96, $F_1$ = 0.84) and led to 11 missed moderate-to-severe OSA patients across all databases (0.2% of all moderate and severe). The learning curve (Fig. S1) of OxiNet performance on the SHHS1 test set demonstrated a monotonous increase as a function of the number of examples in the training set. This illustrates the importance of using a large training set (totaling thousands of recordings) in order to create a robust DL model for our task. Taken together, the results demonstrate the value of OxiNet in reaching performance that may be viable for medical diagnostic use.

The second main contribution of the research was the assessment of the robustness of OxiNet performance by age, sex, ethnicity, and comorbidity. Our experimental results showed that the performance of OxiNet was robust in repetitive measurements (SHHS2, Table 2 and Figs. 1 and 2). Performance dropped when using the external test sets: the drop in performance was significant for a distribution shift relative to ethnicity (CFS) with $F_1$ = 0.66 for the Black and African American participant subgroup against $F_1$ = 0.80 for the white participant subgroup. An overall $F_1$ score of 0.75 was obtained for the MESA database but there was a high variance across the different ethnicities with 0.72 for Hispanic participants, 0.71 for Black and African American participants, 0.78 for white participants and 0.77 for Chinese American participants (Table S3). Thus, while OxiNet performed well on white and Chinese American participants, it performed poorer on Hispanic, Black, and African American participants. These results are consistent with the melanin and typology angle levels used to characterize skin pigmentation in different ethnic groups[18]. It emphasizes the lack of inclusion or low prevalence of such minority groups in datasets traditionally used to train DL algorithms which inevitably leads to embedded biases and poorer diagnostic performance for these groups. Recent research[19,20] has shown that pulse oximeter performance discrepancies have been shown to be affected by patients of different races and ethnicities, leading to poorer clinical management. This may explain the poorer performance of the model on Black and African American participants. The drop in performance in UHV was

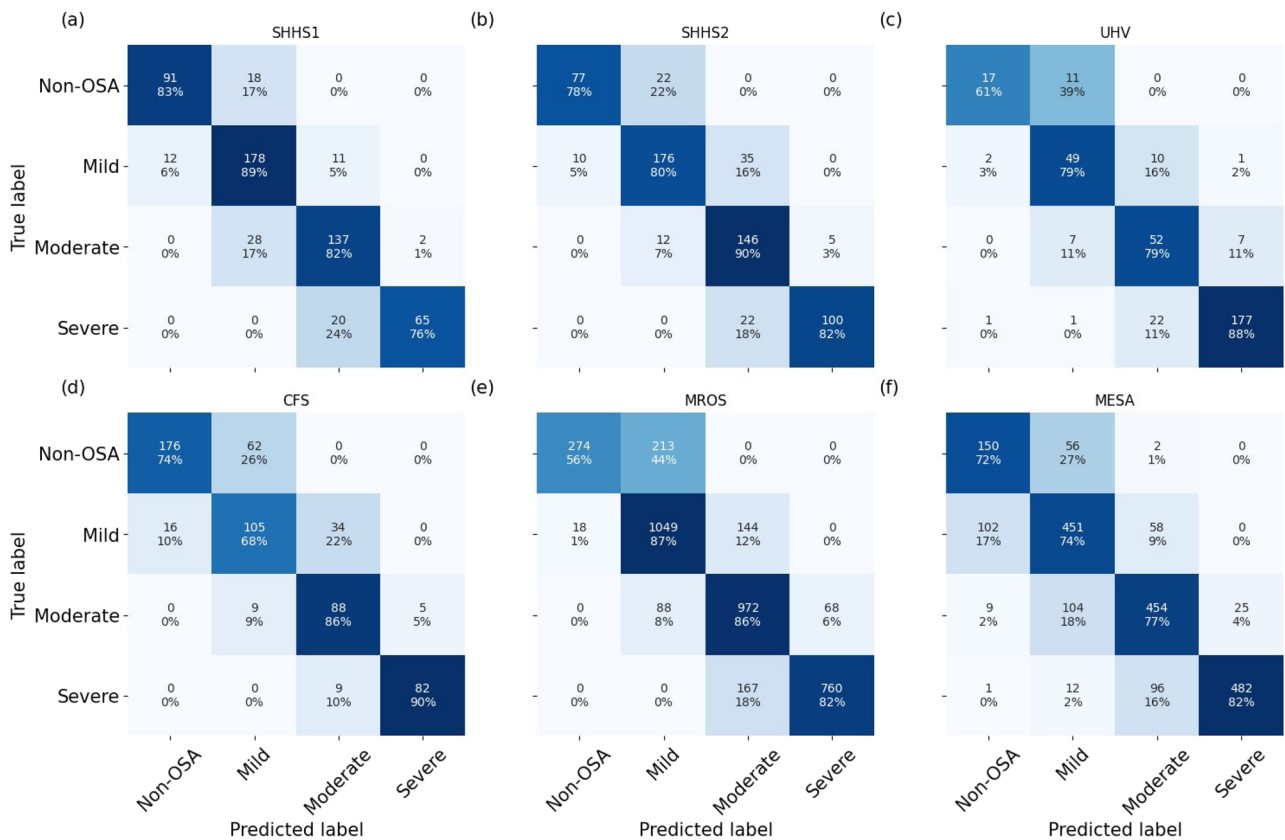

**Fig. 3 | Confusion matrices for OxiNet, for the test set of each external database.** The figure shows the results of OxiNet trained on SHHS1-train: **a** SHHS1; **b** SHHS2; **c** UHV; **d** CFS; **e** MROS; **f** MESA.

due to the presence of COPD comorbidity, which can lead to nocturnal desaturations[13] and then mislead the model. Indeed, 58% of the misclassified patients in UHV had COPD whereas COPD had a prevalence of 20% in this database. Only two severe OSA were classified as non-OSA, in the MESA and UHV databases (0.1% of all 2017 severe cases), as shown in Fig. 3. Only 13 severe OSA cases were classified as mild OSA in the MESA and UHV databases (0.6% of all 2017 severe cases). The two misclassified severe-OSA from UHV had COPD with a global initiative for chronic obstructive lung diseases (GOLD) level three. None of the non-OSA was classified as severe, but an overall of two non OSA from MESA were classified as moderate. These two patients were Black and African American. A total of 15 patients were severe OSA and were classified as non OSA or mild OSA. For these recordings, the TST was <4 h (3.4 ± 0.4). These individuals probably had insomnia, which will affect our approximation $\widehat{\text{TST}}$ of the TST. Overall, the results illustrate that performance is impaired due to important distribution shifts. In particular, we found that distribution shifts due to ethnicity (CFS, MESA) and the presence of significant respiratory comorbidity with COPD (UHV) had an important impact on model performance.

For model explainability, $L_{\text{region}}$ was set to 120 s, which is the order of magnitude of the duration of one to a few apnea events (severe desaturations typically last 30–45 s, ref. [16]). The importance score is calculated as the difference between the predicted AHI on the original signal and the predicted AHI on the signal with the corresponding window replaced by a baseline $SpO_2$ value. In this study, the baseline $SpO_2$ value was set to the mean value of the entire recording. Figure 4 presents examples of three different recordings. Figure 4a displays an overnight signal, where OxiNet identifies clusters of desaturations. OxiNet leverages the temporal context of desaturation events within the overnight time series. This is in opposition to a rule-based ODI detector that searches for desaturations as isolated events, i.e. independent of their temporal context. Figure 4b shows a signal segment

with several apnea events. Although OxiNet assigned relatively high scores, there was no desaturation detected by the rule-based desaturation detector. This reflects that the desaturation detector is too constrained while OxiNet may learn a variety of $SpO_2$ patterns associated with apnea and hypopnea events. Figure 4c shows a segment with no apnea or hypopnea respiratory event and, in agreement, relatively low OxiNet scores. The rule-based ODI detector, however, detected a desaturation that is not associated with a respiratory event. Overall, the explainability figures suggest that OxiNet provides added value over a simpler rule-based desaturation detector. This is because the data-driven approach enables to better learn the representation of SpO2 events during apnea and hypopnea across the high physiological variability of thousands of individuals used to train OxiNet. OxiNet also takes into account the temporal context of events while classical rule based ODI detectors look at an event in an isolated manner.

This study proposed a DL algorithm for estimating the AHI from the oximetry time series and diagnose OSA. The DL OxiNet model outperformed the baseline models in all test databases (Fig. 2). Overall, this large retrospective multicenter study strongly supports the feasibility of single channel oximetry analysis for robust OSA diagnosis. The availability of a robust data driven model using input from a single pulse oximetry sensor may enable large scale diagnosis of OSA while reducing costs and waiting time. It may also enable multiple night testing and thereby even improve OSA diagnosis. Since 2017, the AASM guidelines recommend diagnosing OSA in uncomplicated patients with a single night sleep study[21], which demands high test accuracy. In addition, the test must be low cost, almost fully (if not fully) automated, and, due to the shortage of hospital beds, must enable support testing in the home environment. Within this context, the high performance of OxiNet provide an exciting perspective in enabling remote diagnosis and monitoring of OSA.

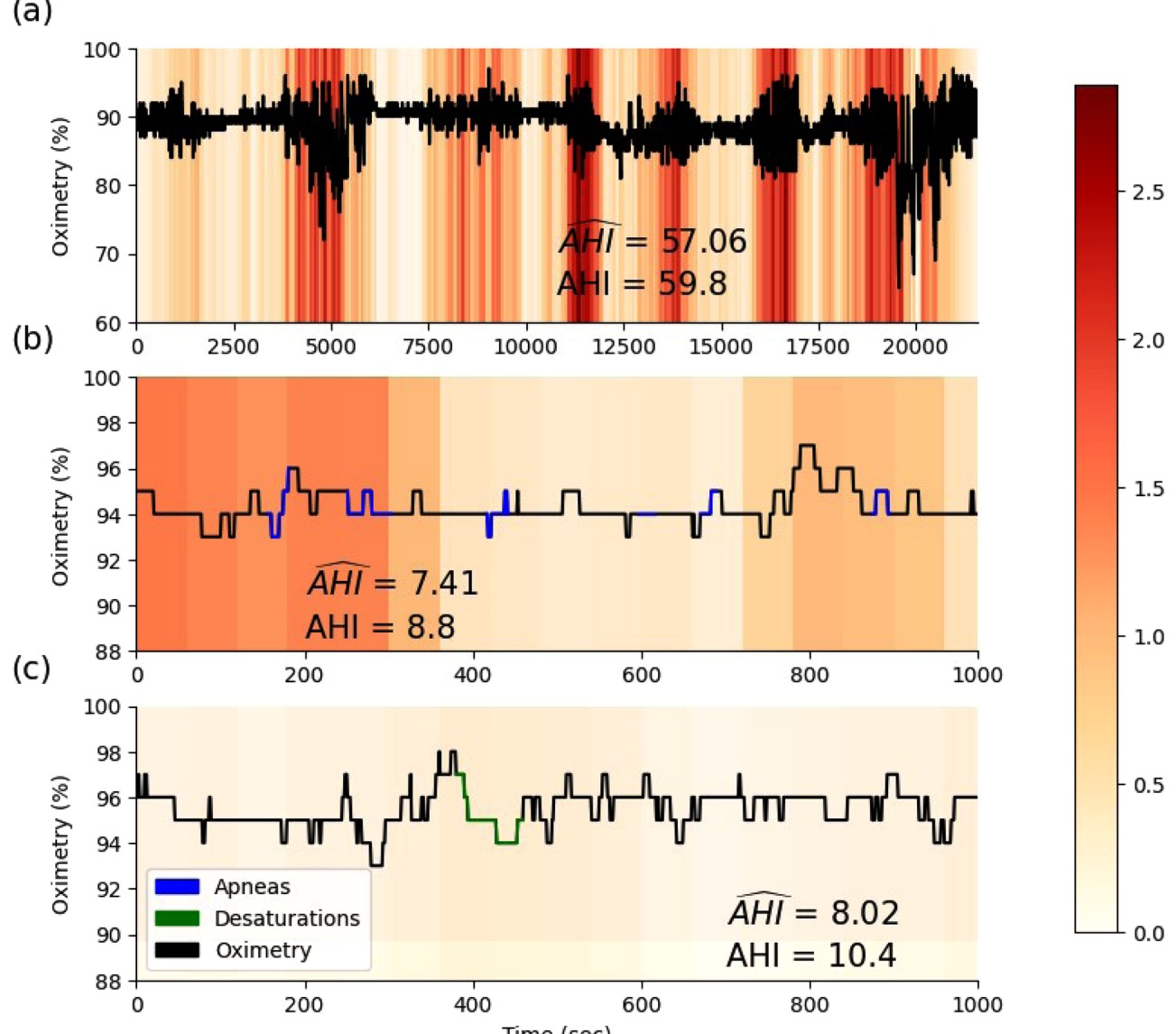

**Fig. 4 | OxiNet explainability.** The three panels display the importance score for sections of three overnight recordings. The panels highlight the particular importance given **a** to desaturations clusters, **b** to apnea events where no desaturation was detected by the rule based desaturation detector, and **c** the low importance given to a section where no apnea or hypopnea event was annotated but where a desaturation was detected by the rule based desaturation detector.

The main limitation of this study is the absence of data from many underrepresented groups, ethnicity, or from developing countries. This is mainly due to the fact that despite the unprecedented initiative of sleepdata.org in open sourcing large databases of sleep data, the majority of the databases hosted on the platform are from the US. To drive health innovation that meets the needs of all and to democratize healthcare, there is a need for more databases from historically underrepresented populations. Beyond the practical and economical aspect of using home oximetry for the diagnosis of OSA, the test could also be repeated for a couple of nights. Indeed, there is evidence that there exists night to night variability of respiratory events in OSA patients[22,23] and that this may lead to misdiagnosis if only a single night test is performed. Thus multiple night testing may be enabled by a home test powered by OxiNet. Some of the databases used in this study were collected at home. This includes the SHHS database. UHV, CFS, MROS and MESA were collected at the hospital. The performance was high for all databases with no particular trend observed between the oximetry data collected at home or at the hospital (see Fig. 2). For this reason, we do not expect much difference in performance given

the usage of a similar clinical grade oximetry device. The same manufacturing company, Nonin, was used for the SHHS, UHV, CFS, MROS, and MESA databases, which were used to assess the performance of OxiNet according to input distribution shifts. Although Nonin has different oximeter products and versions, we believe that since most of the databases were recorded from oximeters from a single manufacturing company then this did not affect significantly the classification outcomes across our test sets. Previous research[24,25] has reported variability in SpO2 measurements across different oximetry devices. Consequently, it will be valuable in future work to report on the performance of OxiNet for different oximeter manufacturers. We approximated the total sleep time ($\widehat{TST}$) as being the time interval between sleep onset and sleep offset. In practice, this can be easily estimated using the photoplethysmography (PPG) signal recorded by the pulse oximeter. In previous research, we have demonstrated the feasibility of sleep staging from PPG, with a kappa score of 0.74 for 4-class classification (wake, light, deep, REM sleep) and an $R$ squared of 0.92 in estimating TST[26]. Because the raw PPG signal was not available for most databases used in this research, we used the sleep stages

provided instead in order to segment the onset and offset of the oximetry signal, as illustrated in Fig. S2. Another limitation of this work is that the datasets used for the analysis are relatively old. SHHS for instance was recorded between 1995 and 1998. Improvements have been made to oximetry technology since then. We do expect that some incremental improvement can be reached provided we had access to a dataset making use of state of the art oximeters and having a similar size to the SHHS. Indeed, although a relatively old dataset, the large size of SHHS was necessary to reach high model performance with OxiNet (Fig. S3). However, we do not expect a change in the relative performance of OxiNet versus the benchmarked models (ODI, OBM) and thus our main conclusions. In this work, we used Gaussian distributed noise for data augmentation. One avenue for further improvement of our approach would be to consider developing a simulator for the purpose of generating more biologically feasible sources of noise that are typical in oximetry measurement.

To conclude, this large, retrospective multicenter analysis provides strong support for the feasibility of single channel oximetry analysis for OSA diagnosis using OxiNet. In addition, it presented an approach to analyzing the performance of a machine learning model to specific population samples. Finally, this research provided a unique example of how large open access databases can enable the assessment of the robustness performance of ML algorithms across ethnicity, age, sex, and comorbidity to ensure the creation of robust and fair ML models.

## Methods
### Databases
The challenge of robustness is often raised in ML, especially for medical applications[27]. Indeed, there are many sources for distribution shifts, here defined as changes in the model input distribution, such as the difference across ethnicity groups. The model could learn a certain "bias" of the training set and generalize poorly on external test sets. We performed a large retrospective multicenter analysis including a total of 12,923 PSG recordings, totaling 115,866 hours of continuous data, from six independent databases. The databases include distribution shifts in age, sex, ethnicity, and comorbidity. The institutional review board from the Technion-IIT Rapport Faculty of Medicine was obtained under number 62-2019 to use the retrospective databases obtained from the open access sleepdata.org resource for this research. Table S4 provides summary statistics on demographics to describe the population samples for each of these databases.

**Sleep Heart Health Study (SHHS).** SHHS[28] is a multicenter cohort study conducted by the National Heart Lung & Blood Institute (ClinicalTrials.gov Identifier: NCT0000527) to determine the cardiovascular and other consequences of sleep-disordered breathing. Individuals were recruited to undergo a type II home PSG. The Nonin XPOD 3011 pulse oximeter (Nonin Medical, Inc., Plymouth, MI, USA) is used for recording. The signal is sampled at 1 Hz. In the first visit, denoted SHHS1, 6441 men and women, aged more than 40 years, are included in the database between November 1, 1995 and January 31, 1998. Recordings from 5793 subjects undergoing unattended full night PSG at baseline are available. A second visit has been performed from January 2001 to June 2003 and will be denoted as SHHS2. This second visit includes 3295 participants.

**Río Hortega University Hospital of Valladolid (UHV).** UHV[29] is composed of 369 oximetry recordings. The original database composed of 350 in lab PSG recordings is further described in Andres Blanco et al. and Levy et al.[13,29]. A total of 19 recordings were added to this research. The Nonin WristOx2 3150 was used to perform portable oximetry (simultaneously to PSG) and sampled at 1 Hz for the first 350 recordings, and 16 Hz for the additional 19. The UHV was the only database that was not part of the National Sleep Research Resource (available on

sleepdata.org). However, the protocol for annotating the UHV PSG recordings also followed the AASM 2012 recommendations[11], and scoring was formed by certified sleep technicians. The UHV database contains patients with suspected sleep disordered breathing and 78 patients with chronic obstructive pulmonary disease (COPD), which is a bias from the other databases. COPD is a lung pathology characterized by persistent airflow limitation that is usually progressive and an enhanced chronic inflammatory response to noxious particles or gases in the airways and the lungs[30].

**Cleveland Family Study (CFS).** The CFS database[31] is made up of 2284 individuals from 361 families, one recording per patient. A subset of the original database, composed of 728 recordings, was available on NSRR and was used for this study. CFS is a large familial sleep apnea study designed to quantify the familial aggregation of sleep apnea. The oximetry was recorded using a Nonin 8000 sensor and sampled at 1 Hz. The database was acquired in the hospital when the patient underwent a type I PSG. Among the 728 recordings available, there are 427 (59%) Black and African American participants.

**Osteoporotic Fractures in Men Study (MROS).** MROS[32] is an ancillary study of the parent Osteoporotic Fractures in Men Study. Between 2000 and 2002, 5994 community dwelling men 65 years or older were enrolled in 6 clinical centers in a baseline examination. Between December 2003 and March 2005, 3135 of these participants were recruited to the sleep study when they underwent a type II home PSG and 3–5-day actigraphy studies. The objectives of the sleep study are to understand the relationship between sleep disorders and falls, fractures, mortality, and vascular disease. The oximetry signal was recorded with a Nonin 8000 sensor and sampled at of 1 Hz.

**Multi Ethnic Study of Atherosclerosis (MESA).** MESA[33] is a six-center collaborative longitudinal investigation of factors associated with the development of sub clinical cardiovascular disease. The study includes PSGs of 2056 individuals divided into four ethnic groups: Black and African American participants ($n = 572$), white participants ($n = 743$), Hispanic participants ($n = 491$), and Chinese American participants ($n = 250$) men and women ages 45–84 years, recorded between 2000 and 2002 with a type II home PSG. The oximetry signals were recorded using a Nonin 8000 sensor, with a sampling rate of 1 Hz.

### Scoring rules
Figure 5 presents the distribution of actual AHI for each database. Table 3 summarizes the base demographic data and the AHI of each database. We ensured that the definitions of apnea and hypopnea events and thus the computation of the reference AHI were homogeneous across databases.

The databases SHHS, CFS, MROS and MESA provided by the NSSR were scored following the procedure described in Redline et al.[28]. Briefly, physiological recordings were originally scored in Compumedics Profusion where apnea and hypopnea respiratory events were scored according to drops in airflow that lasted more than 10 s without criteria of arousal or desaturation for hypopneas. After apneas and hypopneas were identified, the Compumedics Profusion software linked each event to data from the oxygen saturation and EEG channels. This allowed each event to be characterized according to various degrees of associated desaturation and associated arousals and/or combinations of these parameters. This top-down approach enabled the NSSR to generate AHI variables following different recommendations and rules (recommended/alternative). These AHI variables are available on the NSRR website. We used the *ahi_a0h3a* variable, which is consistent with AASM12 recommended rule as specified on the NSSR website and re-confirmed through private correspondence by the NSSR administrators. Data from the UHV were recorded after 2012 and followed the AASM 2012 guidelines. Annotations for all the NSRR

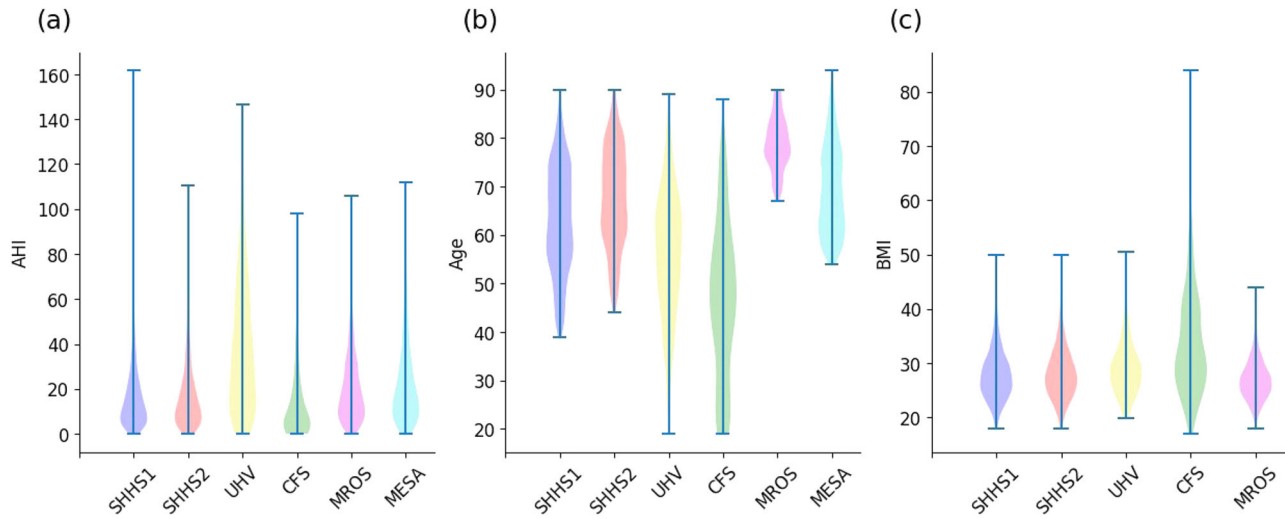

**Fig. 5 | Violin plot.** Violin plot for **a** Apnea Hypopnea Index (AHI), **b** age, and **c** BMI for the study databases. BMI was not available for the MESA database.

**Table 3 | Summary table for all databases used**

| Database | Number | AHI | Age | Male | Timeframe | Main type of shifts |
|---|---|---|---|---|---|---|
| SHHS1 | 5778 | 9.5 ± 15.6 | 63.0 ± 17.0 | 52% | 1995–1998 | – |
| SHHS2 | 621 | 11.3 ± 15.6 | 68.0 ± 16.0 | 54% | 2001–2003 | – |
| UHV | 369 | 33.9 ± 43.8 | 57.0 ± 18.0 | 76% | 2013–2015 | Suspected SDB, COPD |
| CFS | 728 | 4.0 ± 14.9 | 43.0 ± 33.0 | 53% | 2001–2006 | Ethnicity, age |
| MROS | 3937 | 17.0 ± 21.0 | 76.0 ± 5.5 | 100% | 2003–2005 | Men, age |
| MESA | 2056 | 14.3 ± 22.0 | 68.0 ± 14.0 | 46% | 2000–2002 | Ethnicity |

For continuous features, the mean ± standard deviation is presented. The number of recordings is specified after excluding recordings shorter than 4 h. The main type of shifts is provided with respect to the baseline SHHS1 database.
*SDB* sleep-disordered breathing, *COPD* chronic obstructive pulmonary disease.

databases (SHHS, CFS, MROS and MESA) as well as the UHV database were made by certified technicians.

Following the American Academy of Sleep Medicine (AASM) 2012 and ICSD-3 guidelines, the AHI was defined as the average number of apneas and hypopneas per hour of sleep. Apneas were scored if (i) there is a drop in the peak signal excursion by ≥90% of pre-event baseline using an oronasal thermal sensor (diagnostic study), positive airway pressure device flow (titration study), or an alternative apnea sensor; and (ii) the duration of the ≥90% drop in sensor signal is ≥10 s. In the same regard, following the recommended rule, hypopneas were defined as a ≥30% fall in an appropriate hypopnea sensor for ≥10 s and with a ≥3% desaturation or associated arousal.

**Preprocessing and exclusion criteria**
Recordings with technical faults (missing oximetry channel, or corrupted file) and patients under 18 years old, were excluded. Recordings from the UHV database were re-sampled at 1 Hz so that all databases had the same sampling rate. The Delta Filter[34,35] was applied to the oximetry time series, to remove non physiological values due to the motion of the oximeter, or lack of proper contact between the finger and the probe. If there were fewer than three consecutive non-physiological values in the signal, a linear interpolation was performed, to fill in the missing values.

Initiating sleep may take some time and individuals with severe OSA may have numerous overnight awakenings. When computing the AHI in regular PSG examinations, the wake periods are excluded from the computation of the AHI, i.e., the cumulative number of apnea and hypopnea events is divided by the total sleep time. To partially account for this in our experiments, we have defined the sleep onset as the beginning of the first consecutive 5 min segments labeled and sleep

offset as the end of the last consecutive 5 min segments labeled as sleep on the hypnogram provided for each recording. We approximated the total sleep time ($\widehat{TST}$) as being the time interval between sleep onset and sleep offset. In practice, this can be easily estimated using the photoplethysmography (PPG) signal recorded by the pulse oximeter as we have demonstrated in our research work[26].

Signals with $\widehat{TST}<4$ (i.e., less than 4 h of sleep) were excluded[28]. All remaining signals were padded to 7 h. This enables us to handle signals of different lengths, from 4 to 7 h. Patients younger than 18 years were not considered in this study and were removed from the databases.

**Baseline model**
Two baseline classical ML models were implemented to benchmark against the DL approach. The first model included a single oximetry feature, which is the ODI with a threshold at 3%[13]. For the second model, $SpO_2$ features were computed from the oximetry time series using the open source POBM toolbox[35]. These biomarkers are divided into five categories: (1) **General Statistics**: time based statistics describing the oxygen saturation data distribution. For example, Zero-Crossing[36] and delta index[37]. (2) **Complexity**: quantifies the presence of long range correlations in non stationary time series. For example, Approximate Entropy[38], or Detrended Fluctuation Analysis (DFA)[39]. (3) **Periodicity**: quantifies consecutive events to identify periodicity in the oxygen saturation time series. For example, Phase rectified signal averaging (PRSA)[40] and power spectral density (PSD). (4) **Desaturations**: time based descriptive measures of the desaturation patterns occurring throughout the time series. For example, area, slope, length and depth of the desaturations. (5) **Hypoxic Burden**: time-based measures quantifying the overall degree of hypoxemia imposed on the heart and other organs during the recording period. For example,

cumulative time under the baseline (CT)[41]. We have made open source on physiozoo.com the code for computing the digital oximetry features. In addition, the features are more extensively described, including their mathematical definition, in our previous work[35].

A CatBoost regressor[42] was trained, using a total of 178 engineered features. Further description of the features is available in the supplementary note 1. We used the maximum relevance minimum redundancy (mRMR) algorithm for feature selection. The model was optimized with fivefold cross-validation using Bayesian hyper parameter search. For both the ODI and OBM models, sex and age were added as two additional demographic features (Table S4).

## OxiNet
**Architecture.** In contrast to classical ML approaches, DL techniques provide the ability to automatically learn and extract relevant features from the time series. The model takes as input the preprocessed overnight signal. Recordings with a $\widehat{\text{TST}}$ duration of over 4 h were included. Those with a $\widehat{\text{TST}}$ duration of 4−7 h were padded to 7 h. For those with a $\widehat{\text{TST}}$ duration of more than 7 h (less than 3% over all test set examples) only the first 7 h were included. The oximetry signal is independently processed by two branches as inspired by the architecture proposed by Interdonatoa et al.[43]. The first branch is based on convolutions, extracting useful patterns in the time series, and is called Convolutional Neural Network (CNN). The second branch, called Convolutional Recurrent Neural Network (CRNN), exploits the long range temporal correlation present in the time series.

For the CNN branch, the signals were split into overlapping windows of length $L_{\text{window}}$. The windows are fed to the first part of the branch, which extracts local features. This first part is composed of $n_B$ sequential blocks, which extract features from each window. One block is composed of $n_L$ 1D convolutional layers with kernel size 3, batch normalization, and Leaky ReLU activation followed by a maxpool layer of stride 2. There are skipped connections between each block. The local features of each window are then concatenated, and the second part of the model extracts long range temporal features, using dilated convolution. A total of $n_{DB}$ dilated blocks are sequentially used. A dilated block is made of $n_C$ 1D convolutional filters with kernel size $K_{\text{dilated}}$, followed by a Leaky ReLU activation. The dilation rate is progressively increased in order to increase the network's field of view, beginning at rate$_{\text{dilation}}$ and being multiplied by 2 between each block. The CNN branch produces the feature vector $V_{\text{CNN}} \in \mathcal{R}^{N_{\text{CNN}}}$.

For the CRNN branch, a representation of the data is first created, to reduce the temporal resolution. This is done by using 2 CNN blocks when one block is composed of a convolutional layer with kernel size $k_{\text{CRNN}}$, batch normalization, and Leaky ReLu activation followed by a maxpool layer of stride 2. Then, a total of two stacked layers of bidirectional Long Short Term Memory (LSTM) with $n_{\text{LSTM}}$ units is then applied. The CRNN branch produces the feature vector $V_{\text{CRNN}} \in \mathcal{R}^{N_{\text{CRNN}}}$.

The clinical metadata (META) is processed thanks to a fully connected layer, producing the feature vector $V_{\text{META}} \in \mathcal{R}^{N_{\text{META}}}$. The aggregated feature vector $V_{\text{final}} = [V_{\text{CNN}}, V_{\text{CRNN}}, V_{\text{META}}]$ is processed by $n_{\text{classifier}}$ classifier blocks to give the final prediction. A classifier block is composed of a fully connected layer, batch normalization, Leaky Relu activation, and then dropout (with dropout rate $d_{\text{classifier}}$). Each classifier block reduces the dimensionality of the input by 2. A last fully connected layer is then applied, to output the predicted AHI.

This approach allows the model to learn complementary features and better exploit the information hidden in the time series. In order to enforce the discriminating power of the different subsets of features, we adopt the approach proposed by Hou et al.[44]. Two auxiliary regressors were created, working respectively on the $V_{\text{CNN}}$ and $V_{\text{CRNN}}$ vectors. These regressors were not involved in the final prediction of the model, but helped in the training process, by ensuring that each

subset of the features was trained to be independently discriminative. Figure 6 presents the architecture of the resulting OxiNet model.

The experiments were performed on a PowerEdge R740, 1 GPU NVIDIA Ampere A100, 40GB, 512 GB RAM. For our diagnostic objective, evenly distributed errors with low variance are preferable, as they might not change the final diagnosis of the model. For the above reason, the model was optimized using the Mean Squared Error (MSE) loss combined with L2 regularization. A total of two data augmentation techniques are used: moving window and jitter augmentation. More details about the loss and the data augmentation are available in the supplements.

**Loss.** For our diagnostic objective, evenly distributed errors with low variance are preferable, as they might not change the final diagnosis of the model. For the above reason, the model was optimized using the Mean Squared Error (MSE) loss combined with L2 regularization. This loss was computed three times: for the two auxiliary regressors, and for the final prediction. The final loss function used was:

$$\mathcal{L} = \mathcal{L}_{\text{aggregated}} + \lambda_{\text{CNN}} * \mathcal{L}_{\text{CNN}} + \lambda_{\text{CRNN}} * \mathcal{L}_{\text{CRNN}} \tag{1}$$

When $\lambda_{\text{CNN}}, \lambda_{\text{CRNN}}$ are two hyper parameters controlling the impact of $\mathcal{L}_{\text{CNN}}, \mathcal{L}_{\text{CRNN}}$, respectively. At the beginning of the training process, $\lambda_{\text{CRNN}} = \lambda_{\text{CNN}} = 1$. Then every four epochs the two hyper parameters are multiplied by 80%, so the weight of the auxiliary classifiers in the final loss decreases. The intuition is that these regressors help the final model to converge, but are not part of it. That is why as long as the training process continues, their weight is decaying.

**Data augmentation.** The OxiNet model is composed of approximately 870,000 parameters, which is a few orders of magnitude larger than the number of examples contained in our training set. Data augmentation was used to increase the training set size, especially Jitter augmentation, adding white noise to the signal. The generated signal is:

$$X_{\text{new}} = X + N, \quad N \sim \mathcal{N}(0, \sigma_{\text{noise}}) \tag{2}$$

where $X_{\text{new}}$ is the signal generated, $X$ is the original signal and $N$ is the noise added. $\sigma_{\text{noise}}$ is a hyperparameter of the model. Figure S4 presents the original and generated signals, with $\sigma_{\text{noise}} = 0.5$. Although the generated signal may not be biologically feasible, this augmentation technique adds variance to the samples that are fed to the model and prevent overfitting the training set.

## Training strategy
The SHHS1 database was split into a 90% training set and a 10% test set. All the hyperparameters of the model were optimized using Bayesian search, over 100 iterations. To this end, the SHHS1 training set was split into 70% training and 30% validation. In the first step, the model was trained on the SHHS1 train for 100 epochs. The Adam optimization algorithm was used, with a learning rate of 0.005. The set of hyperparameters leading to the smallest validation loss was retained. Then the performance measures were reported for the test set for each database, independently.

## Explainability
Explainability is a critical aspect to ensure that the model is trustworthy and can be integrated into clinical practice. It enables the identification of the contributing factors and provides explanations for the predictions made by the model. Indeed, DL models are known for their black box nature, making it difficult to understand how they arrive at their predictions. To that end, we adapted the algorithm proposed by Zeiler et al.[45], named Feature Occlusion (FO) and originally proposed for image recognition. The algorithm has already been

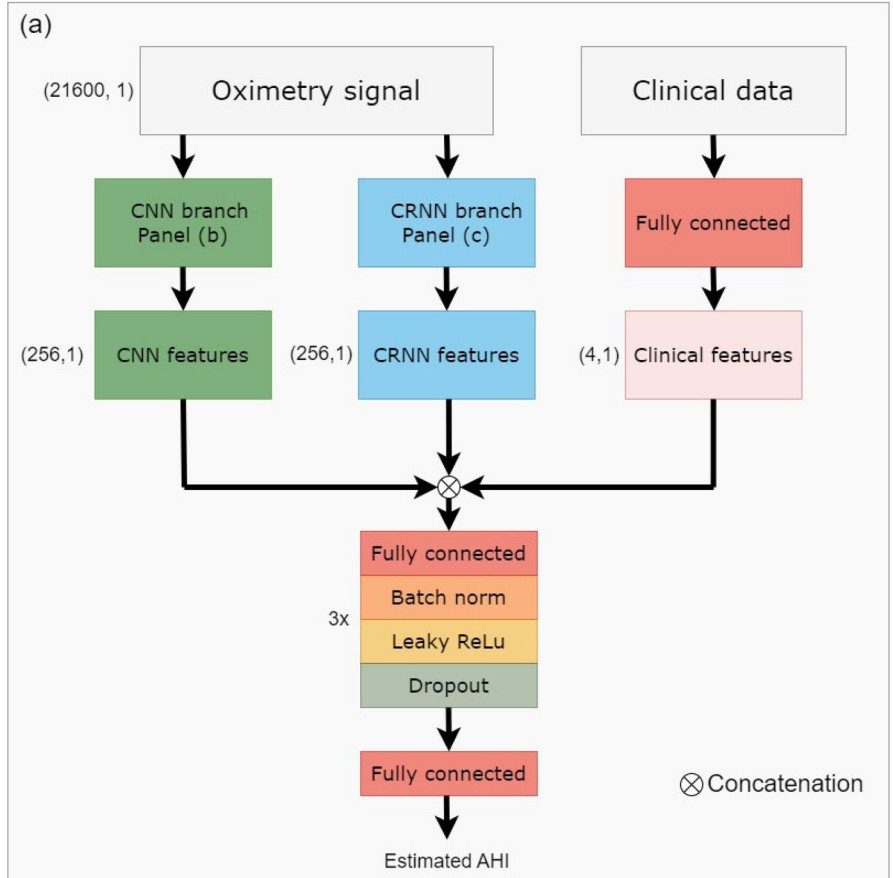

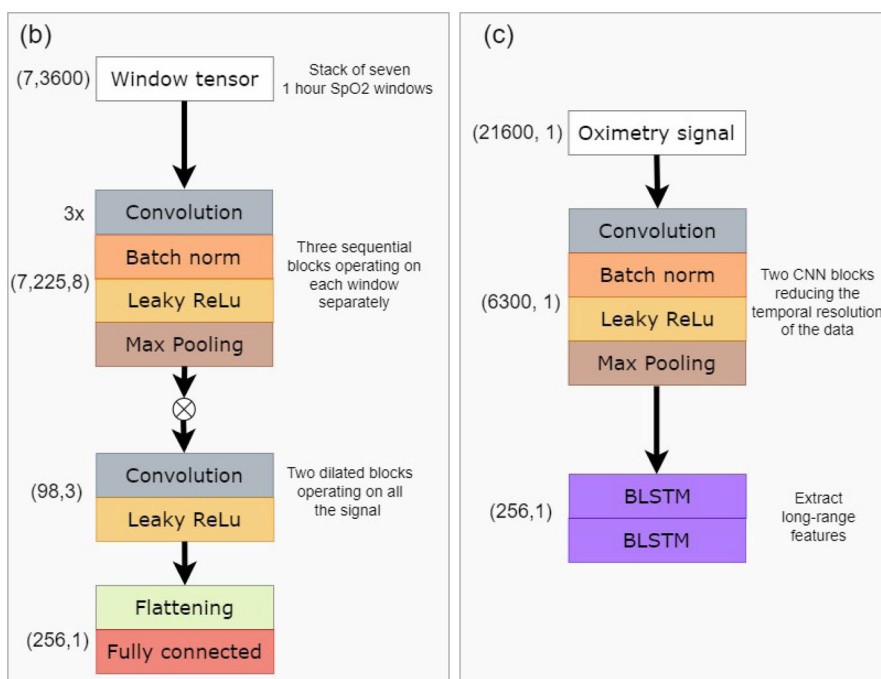

**Fig. 6 | OxiNet architecture. a** Shows a high-level overview of the overall architecture. The raw data is independently processed by a CNN branch and a CRNN branch. The concatenation of CNN, CRNN, and clinical features is processed by a regressor that estimates the AHI. **b** Shows more in detail the CNN branch, while **c** presents the CRNN branch. BLSTM bidirectional long short term memory, CNN convolutional neural network, CRNN convolutional recurrent neural network.

used in the context of time series prediction several times[46,47]. The algorithm computes the importance score as the difference in output after replacing each contiguous region with a given baseline. We defined a region as a window of $L_{region}$ seconds in the oximetry signal and performed the occlusion with a sliding window of size $L_{region}/2$, in order to have an importance score for each batch of $L_{region}/2$ seconds. The baseline to replace with was set to be the overall mean of the signal.

## Performance measures

The Kruskal–Wallis test was applied ($p$ value cut-off of 0.05) to evaluate whether individual demographic features were discriminating between the four groups of OSA severity: non OSA ($AHI < 5$), mild ($5 \leq AHI < 15$), moderate ($15 \leq AHI < 30$) or severe ($AHI \geq 30$) OSA. Table S4 presents the summary statistics for these variables and across the databases. Bland-Altman and correlation plots were generated to analyze the agreement and association between the estimated and reference AHI. The agreement was displayed as the median difference between $\widehat{AHI}$ and AHI and the 5th and 95th percentiles of their difference. For the regression task, the Intraclass Correlation Coefficient (ICC) was reported and is defined as follows:

$$ICC = \frac{MS_I - MS_E}{MS_I + (O - 1)MS_E + O * \frac{MS_O - MS_E}{n}} \tag{3}$$

where $O$ is the number of observers (two, in this case, the real and predicted AHI), $MS_I$ is the instances mean square, $MS_E$ is the mean square error and $MS_O$ is the observers mean square.

After converting the AHI into the four levels of severity (i.e., non-OSA, mild, moderate, and severe OSA), the macro averaged $F_1$ score was reported as the measure of diagnostic accuracy. The $F_1$ score was computed as follows:

$$Se_M = \frac{1}{4} \sum_{k=1}^{4} \frac{TP_k}{TP_k + FN_k} \tag{4}$$

$$PPV_M = \frac{1}{4} \sum_{k=1}^{4} \frac{TP_k}{TP_k + FP_k} \tag{5}$$

$$F_{1,M} = 2 \frac{PPV_M * Se_M}{PPV_M + Se_M} \tag{6}$$

where, for a given class $k$, $TP_k$ is the number of true positives, $TN_k$ the number of true negatives, $FP_k$ the number of false positives, and $FN_k$ the number of false negatives. Additional performance measures are defined in the Supplementary Note.

We estimated the confidence interval for the $F_1$ and ICC scores of the different models using bootstrapping, similar to the work of Biton et al.[48]. That is, the $F_1$ and ICC scores were repeatedly computed on randomly sampled 80% of the test set (with replacement). The procedure was repeated 1000 times and used to obtain the intervals, which are defined as follows:

$$C_n = \bar{x} \pm z_{0.95} * se_{boot}$$

with $\bar{x}$ as the bootstrap mean, $z_{0.95}$ is the critical value found from the distribution table of normal CDF, and $se_{boot}$ is the bootstrap estimate of the standard error. Bootstrap was performed on each database separately. To determine if there was a statistical difference, the Wilcoxon rank-sum test was applied and a $p$ value cut-off at 0.05 was used. The statistical test was also used to determine if there is a significant difference in performance measures for male vs female.

## Data availability

The databases SHHS, CFS, MROS, and MESA are been achieved by the National Sleep Research Resource with appropriate deidentification. Permission and access for accessing these datasets were obtained via the online portal: https://www.sleepdata.org. In addition, the UHV database was contributed by co-author Prof. Felix Del Campo (fsas@telefonica.net) and access may be obtained on request.

## Code availability

Our code and experiments can be reproduced by utilizing the details provided in the "Methods" section. For OxiNet, this includes the model architecture (section "Architecture" and Fig. 6), loss function (section "Loss"), data augmentation (section "Data augmentation"). Our trained model is also available at https://github.com/jeremy-levy/OxiNet/tree/main and is provided for academic research purpose and under a GNU GPL license. The source code used to engineer the oximetry biomarkers and train the ODI and OBM models has been made available at (https://oximetry-toolbox.readthedocs.io/en/latest/). Adam initialization was used https://www.tensorflow.org/api_docs/python/tf/keras/optimizers/Adam). For the data augmentation, we used the GaussianLayer (https://www.tensorflow.org/api_docs/python/tf/keras/layers/GaussianNoise). The code for explainability is adapted from the work of Zeiler et al.[45] and is available at https://github.com/saketd403/Visualizing-and-Understanding-Convolutional-neural-networks.

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

## Acknowledgements

The Sleep Heart Health Study (SHHS) was supported by National Heart, Lung, and Blood Institute cooperative agreements U01HL53916 (University of California, Davis), U01HL53931 (New York University), U01HL53934 (University of Minnesota), U01HL53937 and U01HL64360 (Johns Hopkins University), U01HL53938 (University of Arizona), U01HL53940 (University of Washington), U01HL53941 (Boston University), and U01HL63463 (Case Western Reserve University). The National Sleep Research Resource was supported by the National Heart, Lung, and Blood Institute (R24 HL114473, 75N92019R002). The Cleveland Family Study (CFS) was supported by grants from the National Institutes of Health (HL46380, M01 RR00080-39, T32-HL07567, RO1-46380). The National Sleep Research Resource was supported by the National Heart, Lung, and Blood Institute (R24 HL114473, 75N92019R002). The National Heart, Lung, and Blood Institute provided funding for the ancillary MrOS Sleep Study, "Outcomes of Sleep Disorders in Older Men," under the following grant numbers: R01 HL071194, R01 HL070848, R01 HL070847, R01 HL070842, R01 HL070841, R01 HL070837, R01 HL070838, and R01 HL070839. The National Sleep Research Resource was supported by the National Heart, Lung, and Blood Institute (R24 HL114473, 75N92019R002). The Multi-Ethnic Study of Atherosclerosis (MESA) Sleep Ancillary study was funded by NIH-NHLBI Association of Sleep Disorders with Cardiovascular Health Across Ethnic Groups (RO1 HL098433). MESA is supported by NHLBI funded contracts HHSN268201500003I, N01-HC-95159, N01-HC-95160, N01-HC-95161, N01-HC-95162, N01-HC-95163, N01-HC-95164, N01-HC-95165, N01-HC-95166, N01-HC-95167, N01-HC-95168 and N01-HC-95169 from the National Heart, Lung, and Blood Institute, and by cooperative agreements UL1-TR-000040, UL1-TR-001079, and UL1-TR-001420 funded by NCATS. The National Sleep Research Resource was supported by the National Heart, Lung, and Blood Institute (R24

HL114473, 75N92019R002). J.A.B. and J.L. acknowledge the financial support of Israel PBC-VATAT and by the Technion Center for Machine Learning and Intelligent Systems (MLIS). D.Á. is supported by a "Ramón y Cajal" grant (RYC2019-028566-I) from the "Ministerio de Ciencia e Innovación - Agencia Estatal de Investigación" co-funded by the European Social Fund and in part by Sociedad Española de Neumología y Cirugía Torácica (SEPAR) under project 649/2018 and by Sociedad Española de Sueño (SES) under the project "Beca de Investigación SES 2019. In addition, D.Á. has been partially supported by "CIBER en Bioingeniería, Biomateriales y Nanomedicina (CIBERBBN)" through "Instituto de Salud Carlos III" co-funded with FEDER funds.

## Author contributions

J.A.B. conceived and designed the research, J.L. created OxiNet and cured and analyzed the data. D.Á. and F.C.M. provided statistical and clinical guidance on the interpretation of the results. J.A.B. and J.L. drafted the manuscript; J.L. prepared the figures; J.A.B., J.L., D.Á., and F.C.M. edited and revised the manuscript and approved the final version.

## Competing interests

J.A.B. holds shares in SmartCare Analytics Ltd. The other authors declare no competing interests.
