## [Peer Review File · Nature Communications]

Deep learning for obstructive sleep apnea diagnosis based on single channel oximetryReviewers' comments:

Reviewer #1 (Remarks to the Author):

The authors develop a novel deep learning approach for estimating AHI, which has been trained on a large dataset with given AHI values and validated on various other datasets. Results show it outperforms the baselines with quite a margin.

However, I feel the title is misleading in the sense that generalizability would imply that it works on all datasets, while it only seems to perform well after transfer learning. This makes you wonder: are AHI ratings different across datasets? What are the distribution shifts and the concept drifts in this application? Also exclusion criteria during data selection contradict the notion of a "generalisable" model?

On the model, there is only a high level overview, which makes it impossible to understand the actual architecture or replicate the results. In particular, it is unclear what the window length is that is analysed (7 hours?, but the figure S7 shows 1800 seconds??)

Reviewer #2 (Remarks to the Author):

The authors report the development of a deep learning model to analyze pulse oximetry with a view towards classifying patients with obstructive sleep apnea. The topic is of interest.

Major comments:

1. it seems the AHI criteria differ across the various cohorts. As such it is not clear what is meant by generalizable when the authors use this term. There is transfer learning to make it work with different cohorts suggesting it would not be easily deployable in a new setting. Can the authors provide details about when the model could be used 'out of the box' vs. needing fine tuning? what are the criteria to decide when the model is adequately tuned ? were there stopping criteria ? how many iterations ? How much of this fine tuning is a result of the differing criteria and equipment used in the various sleep cohorts?
2. there are quite a few grammatical errors throughout making it hard to read the manuscript
3. the amount of data excluded for various reasons seems quite substantial, perhaps limiting the applicability of the techniques. Given that skin color may be an important finding, are these patients preferentially excluded ? are there any other characteristics of the excluded patients that should be noted?
4. Why is Gaussian/normally distributed noise is a good way of generating augmented data? Is there a way to add more biologically feasible sources of noise? For instance, in the ECG processing domain muscle artifacts, baseline wander, and other types of non-white noise are often used.
5. Please use statistical tests to show the improvement in model performance is statistically significant. it would be helpful to see ROC curves with the incremental improvement with various models e.g. what is the AUC from standard analyses of oximetry vs. the incremental improvement with DL method ?

Reviewer #3 (Remarks to the Author):

This paper by Levy et al. represents an effort to assess the prediction performance of single-channel oximetry for OSA using deep learning. The diagnosis of OSA is expensive, and numerous patients remain undiagnosed. Out-of-lab testing has value in reducing costs, preventing morbidity, and improving healthcare access. In this context, this paper represents an effort to rationalize the application of single-channel oximetry to a strategy of at-home testing. The results are exciting but there are some major concerns, especially concerning the derivation of the dependent variable. Here are the main points:

Major weaknesses

1. I am concerned about the dependent variable AHI, and its variable definitions that need to be clarified in the paper. They mention the 2012 scoring criteria used for all datasets in the paper. At

first glance, it seemed as if the scoring was provided as a variable in the repository, but it is unclear how this was scored. Who provided the scoring? Did the authors obtain the traces and re-score the events?

For example, the SHHS1 scoring criteria

(https://sleepdata.org/datasets/shhs/files/m/browser/documentation/SHHS1_Protocol.pdf?inline=1) are shown in the accompanying manual as '25% or more reduction of airflow from baseline', and hypopneas as '70% or more reduction of airflow' (along with standard desaturation and arousal thresholds). This differs from the 2012 criteria that scored an apnea as > 90% or more reduction in airflow from baseline, and hypopnea as > 30% reduction in airflow along with other desaturation and arousal thresholds. The authors state that 2012 definitions were 'harmonized' across the dataset. This is particularly relevant to the SHHS dataset which is nearly 25 years old and questions remain about the application of 2012 criteria to that dataset.

2. This study needs to detail the implications of using data of which > 50% comprises recordings over two decades old. In contrast to what the authors describe as the similarities between the various devices, oximeter technologies have evolved.

3. These issues tied to the change in datasets over time lead to the obvious question: what is common among these recordings that align with a diagnosis of OSA or its severity? What is the interpretation of these results that will advance the understanding of the underlying pathophysiology? The need for interpretability is concerning.

4. Many studies have demonstrated the utility of machine learning methods for predicting OSA severity from oximeter tracings. While there is merit in establishing the generalizability of these predictions across multiple datasets, this is an incremental advance over the previous literature in the field.

Minor weaknesses:

1. Abstract: The authors skip over the relevance of home tests. Recognizing that the current home tests provide acceptable diagnostic performance is essential. Many components (e.g., nasal airflow) provide additional information in the possible attribution of localization of obstruction, or for sleep staging, which is not provided by oximetry.

2. The comparison with non-clinical portable oximeters is irrelevant as regulatory organizations usually require these devices to have acceptance criteria before approving them for home use.

3. The authors scored the UHV dataset in-house, which is acceptable, but the potential bias should be acknowledged.

OxiNet: a robust deep learning model for obstructive sleep apnea diagnosis based on single channel oximetry

Jeremy Levy, Daniel Álvarez, Félix Del Campo and Joachim A. Behar

Dear Reviewers,

We thank the reviewers for their appreciation of our work as well as for their feedback which have helped further improve the manuscript significantly.

In this new version of the manuscript, the main modifications that were made include:

- We modified the set of databases we are experimenting with and ensured that the AHI variable used was consistent with the AASM12 recommended rule as specified on the National Sleep Research Resource (NSSR) website. This was re-confirmed through private correspondence by the NSSR administrators operating at Harvard Division of Sleep Medicine. Table 1 summarizes the databases that are now considered for analysis. All the result tables, figures and discussion have been updated accordingly.
- Our exclusion criteria have been revised and now only consist in very minimalistic criteria: technical fault, children (age<18 years old) and <4h of total sleep time. As a result, only a small percentage of recordings from the original datasets are now excluded. This is reflected in Table 2.
- We focus the manuscript on the elaboration of OxiNet and the evaluation of its generalization performance across distribution shifts. Thus, we removed the transfer learning experiment and removed the WSC database simulating a concept shift.
- We added a new section on the explainability of the OxiNet model as well as visualizations (Figure 5).

In addition, we have addressed all the questions that were raised by the reviewers and answered their remarks point by point. We will be happy to clarify any further points.

Sincerely,

Joachim A. Behar, corresponding author.

Reviewer #1 (Remarks to the Author):

The authors develop a novel deep learning approach for estimating AHI, which has been trained on a large dataset with given AHI values and validated on various other datasets. Results show it outperforms the baselines with quite a margin.

We thank the reviewer for his/her appreciation of our manuscript as well as the additional comments. We have tried to answer them in the best manner and will be happy to clarify any further questions.

However, I feel the title is misleading in the sense that generalizability would imply that it works on all datasets, while it only seems to perform well after transfer learning. This makes you wonder: are AHI ratings different across datasets? What are the distribution shifts and the concept drifts in this application? Also exclusion criteria during data selection contradict the notion of a "generalisable" model?

Generalization performance

Following the reviewer's comments and in order to make the focus of our manuscript clearer we have decided to remove the transfer learning experiment and focus the work on the elaboration of OxiNet and the evaluation of its generalization performance relative to state-of-the-art approaches. As the reviewer suggested we removed the word "generalization performance" from the title in order to avoid the confusion with the corresponding machine learning field. The new title is: "OxiNet: a robust deep learning model for obstructive sleep apnea diagnosis based on single channel oximetry". We add that although generalization performance approaches may well be used to somehow boost some more the performance of OxiNet on the external datasets, we believe that this improvement would be limited. Rather, we believe, the remaining gap that exists between OxiNet performance on the external test sets and the local test set is very likely due to Human inter rater variability in annotating apnea and hypopnea events. In a previous study by Magalang UJ et. al. Sleep. 2013 Apr 1;36(4):591-6, the authors reported a human inter rater variability of ICC=0.95 for AHI estimation. In our work, the ICC reported on the external datasets was in the ICC range of 0.92-0.94 which is close from human inter rater variability.

Reference AHI across datasets and scoring rules:

We modified the set of databases and ensured that the AHI variable used was consistent with AASM12 recommended rule as specified on the NSSR website. This was re-confirmed through private correspondence with the NSSR administrators. We have added an extensive section "Scoring Rules" that provides the necessary information.

Distribution and concept shifts:

The main distribution shifts that exist across the different databases are listed in Table 1 and include ethnicity, age, sex and comorbidities. Within the new experimental settings there are no concept shifts.

Exclusion criteria

We have updated our exclusion criteria in order to be as inclusive as possible. The updated exclusion criteria are: corrupted files (which are not usable), patients under 18 (this research focuses on OSA diagnosis in adults) and recordings with less than 4 hours of total sleep time (the standard in remote sleep tests). In addition, to avoid any information leakage in reporting performance on SHHS2 we

made sure to remove individuals who were included in the SHHS1 training set. The update tables reflects that the large majority of the recordings are now kept in the test set databases.

Database	Original number of recordings	Technical fault	$\widehat{TST} < 4$	Age < 18	Recordings after exclusion criteria
SHHS1	5,778	59 (1%)	99 (2%)	0 (0%)	5,620 (97%)
SHHS2	621	0 (0%)	16 (3%)	0 (0%)	605 (97%)
UHV	369	0 (0%)	12 (3%)	0 (0%)	357 (97%)
CFS	728	13 (2%)	71 (10%)	58 (8%)	586 (80%)
MROS	3,937	51 (1%)	133 (3%)	0 (0%)	3,753 (95%)
MESA	2,056	0 (0%)	54 (3%)	0 (0%)	2,002 (97%)

Table 2: Size of each database before and after applying the exclusion criteria. 74% of SHHS2 were removed, because the patients from SHHS1-train were excluded to avoid any leakage information.

On the model, there is only a high level overview, which makes it impossible to understand the actual architecture or replicate the results. In particular, it is unclear what the window length is that is analysed (7 hours?, but the figure S7 shows 1800 seconds??)

We agree with the reviewer that the model description was missing clarify. We re-wrote the model description section in its entirety and produced a clearer figure (Figure 2) summarizing the model architecture. We believe that the updated description and figure provides all the necessary information to reproduce our results. Furthermore, we have specified in the figure S7 (which is now S4) that only a subset of the whole signal is represented, to appreciate the different patterns existing in the signal.

Figure 2: OxiNet architecture. Panel (a) shows a high-level overview of the overall architecture. The raw data is independently processed by a CNN branch and a CRNN branch. The concatenation of CNN, CRNN, and clinical features is processed by a regressor that estimates the AHI. Panel (b) shows more in detail the CNN branch, while panel (c) presents the CRNN branch. BLSTM: bidirectional long short term memory, CNN: convolutional neural network, CRNN: convolutional recurrent neural network.

Reviewer #2 (Remarks to the Author):

The authors report the development of a deep learning model to analyze pulse oximetry with a view towards classifying patients with obstructive sleep apnea. The topic is of interest.

We thank the reviewer for his/her appreciation of our manuscript as well as the additional comments. We have tried to answer them in the best manner and will be happy to clarify any further questions.

Major comments:

1. it seems the AHI criteria differ across the various cohorts. As such it is not clear what is meant by generalizable when the authors use this term. There is transfer learning to make it work with different cohorts suggesting it would not be easily deployable in a new setting. Can the authors provide details about when the model could be used 'out of the box' vs. needing fine tuning? what are the criteria to decide when the model is adequately tuned ? were there stopping criteria ? how many iterations ? How much of this fine tuning is a result of the differing criteria and equipment used in the various sleep cohorts?

Reference AHI across datasets and scoring rules:

Following the reviewer's comments, we decided to modify the set of databases used for the experiments. We ensured that the AHI variable used was consistent with AASM12 recommended rule as specified on the NCSR website. This was re-confirmed through private correspondence with the NCSR administrators. We have added an extensive section "Scoring Rules" that provides the necessary information.

Generalization performance

Following the reviewer's comments and in order to make the focus of our manuscript clearer we have decided to remove the transfer learning experiment and focus the work on the elaboration of OxiNet and the evaluation of its generalization performance relative to state-of-the-art approaches.

2. there are quite a few grammatical errors throughout making it hard to read the manuscript

We thank the reviewer for his/her remark. We went over the manuscript again to fix the grammatical errors.

3. the amount of data excluded for various reasons seems quite substantial, perhaps limiting the applicability of the techniques. Given that skin color may be an important finding, are these patients preferentially excluded ? are there any other characteristics of the excluded patients that should be noted?

We have updated our exclusion criteria in order to be as inclusive as possible. The updated exclusion criteria are: corrupted files (which are not usable), patients under 18 (this research focuses on OSA diagnosis in adults) and recordings with less than 4 hours of total sleep time (the standard in remote sleep tests). In addition, to avoid any information leakage in reporting performance on SHHS2 we made sure to remove individuals who were included in the SHHS1 training set. The update tables reflects that the large majority of the recordings are now kept in the test set databases. Individuals with dark skin colors were not preferentially excluded.

Database	Original number of recordings	Technical fault	$\widehat{TST} < 4$	Age < 18	Recordings after exclusion criteria
SHHS1	5,778	59 (1%)	99 (2%)	0 (0%)	5,620 (97%)
SHHS2	621	0 (0%)	16 (3%)	0 (0%)	605 (97%)
UHV	369	0 (0%)	12 (3%)	0 (0%)	357 (97%)
CFS	728	13 (2%)	71 (10%)	58 (8%)	586 (80%)
MROS	3,937	51 (1%)	133 (3%)	0 (0%)	3,753 (95%)
MESA	2,056	0 (0%)	54 (3%)	0 (0%)	2,002 (97%)

Table 2: Size of each database before and after applying the exclusion criteria. 74% of SHHS2 were removed, because the patients from SHHS1-train were excluded to avoid any leakage information.

4. Why is Gaussian/normally distributed noise is a good way of generating augmented data? Is there a way to add more biologically feasible sources of noise? For instance, in the ECG processing domain muscle artifacts, baseline wander, and other types of non-white noise are often used.

Indeed, the gaussian distributed noise is not necessarily a biological source of noise. However, it allows to add variances to the samples and prevents over-fitting of the model to the training set. We have added the following sentence in the sub-section about data augmentation:

“Although the generated signal may not be biologically exact, this augmentation technique adds variance to the samples that are fed to the model and prevent overfitting.”

The reviewer is correct that more physiological/realistic sources of noise could be added similar to what is done in the ECG domain where simulated baseline wander, muscle artifacts etc. can be used to augment the data. Although common in ECG analysis, such physiological simulators are, to our knowledge, less standard in oximetry analysis. We have, however, added the following sentence to the discussion in order to stress that this could be a valuable avenue for further research:

“In this work we used Gaussian distributed noise for data augmentation. One avenue for further improvement of our approach would be to consider developing simulator for more biologically feasible sources of noise that are typical in oximetry measurement.”

5. Please use statistical tests to show the improvement in model performance is statistically significant. it would be helpful to see ROC curves with the incremental improvement with various models e.g. what is the AUC from standard analyses of oximetry vs. the incremental improvement with DL method?

Following the reviewer’s comment, we have added statistical tests to show that the improvement of OxiNet over different baselines is statistically significant. This was done using bootstrap. We also added statistical test to determine if there was a significant difference in performance measures between male and female. We have added the following sentences in the “Performance measures” section, which we renamed “Performance measures and statistical tests”.

“We estimated the confidence interval for the F1 score of the different models compared using bootstrapping. That is, the F1 score was repeatedly computed on randomly sampled 80% of the test set (with replacement). The procedure was repeated 1000 times and used to obtain the intervals, which are defined as follows:

$$C_n = \bar{x} \pm z_{0.95} * se_{boot}$$

Where \bar{x} is the bootstrap mean, $z_{0.95}$ is the critical value found from distribution table of normal CDF, and se_{boot} is the bootstrap estimate of the standard error. Bootstrap is performed on each database separately. To determine if there was a statistical difference, the Wilcoxon rank-sum test was applied and a p-value cut-off at 0.05 was used. The statistical test was also used to determine if there is a significant difference in performance measures for male vs female.”

We have consequently modified the following table in the “Results” section:

	ODI		OBM		OxiNet	
	ICC	F_{1,M}	ICC	F_{1,M}	ICC	F_{1,M}
SHHS1	0.89 (0.89-0.93)	0.69 (0.68-0.72)	0.93 (0.90-0.95)	0.74 (0.70-0.78)	0.96 (0.95-0.97)	0.84 (0.82-0.86)
SHHS2	0.89 (0.87-0.91)	0.69 (0.65-0.72)	0.93 (0.92-0.93)	0.74 (0.74-0.75)	0.95 (0.97-0.99)	0.83 (0.83-0.85)
UHV	0.75 (0.74-0.78)	0.61 (0.58-0.62)	0.86 (0.88-0.92)	0.67 (0.60-0.74)	0.92 (0.94-0.96)	0.77 (0.77-0.79)
CFS	0.7 (0.66-0.78)	0.56 (0.50-0.70)	0.75 (0.68-0.80)	0.6 (0.51-0.69)	0.92 (0.9-0.96)	0.78 (0.74-0.82)
MROS	0.7 (0.68-0.72)	0.52 (0.50-0.58)	0.81 (0.79-0.83)	0.65 (0.62-0.68)	0.94 (0.95-0.99)	0.80 (0.78-0.84)
MESA	0.75 (0.71-0.77)	0.6 (0.54-0.64)	0.75 (0.70-0.80)	0.65 (0.62-0.68)	0.94 (0.92-0.94)	0.75 (0.72-0.76)

We have added the following sentences in the Results section:

“P-values of Wilcoxon rank-sum tests between OxiNet and OBM were below 0.05 for all databases. The statistical test did not show any significant difference in the performance measures of the model between males and females.”

Regarding the ROC curves: Standard ROC curve is adequate for a binary classification task. In our experiments we perform a regression of the AHI and a subsequent classification into 4 classes: non-OA, mild, moderate and severe OA. Accordingly, we have reported classical performance measures for regression and multiclass classification tasks in Table S3 and S4.

Reviewer #3 (Remarks to the Author):

This paper by Levy et al. represents an effort to assess the prediction performance of single-channel oximetry for OSA using deep learning. The diagnosis of OSA is expensive, and numerous patients remain undiagnosed. Out-of-lab testing has value in reducing costs, preventing morbidity, and improving healthcare access. In this context, this paper represents an effort to rationalize the application of single-channel oximetry to a strategy of at-home testing. The results are exciting but there are some major concerns, especially concerning the derivation of the dependent variable. Here are the main points:

We thank the reviewer for his/her appreciation of our manuscript as well as the additional comments. We have tried to answer them in the best manner and will be happy to clarify any further questions.

Major weaknesses

1. I am concerned about the dependent variable AHI, and its variable definitions that need to be clarified in the paper. They mention the 2012 scoring criteria used for all datasets in the paper. At first glance, it seemed as if the scoring was provided as a variable in the repository, but it is unclear how this was scored. Who provided the scoring? Did the authors obtain the traces and re-score the events? For example, the SHHS1 scoring criteria (https://sleepdata.org/datasets/shhs/files/m/browser/documentation/SHHS1_Protocol.pdf?inline=1)

are shown in the accompanying manual as '25% or more reduction of airflow from baseline', and hypopneas as '70% or more reduction of airflow' (along with standard desaturation and arousal thresholds). This differs from the 2012 criteria that scored an apnea as > 90% or more reduction in airflow from baseline, and hypopnea as > 30% reduction in airflow along with other desaturation and arousal thresholds. The authors state that 2012 definitions were 'harmonized' across the dataset. This is particularly relevant to the SHHS dataset which is nearly 25 years old and questions remain about the application of 2012 criteria to that dataset.

Following the reviewer's comments, we decided to modify the set of databases used for the experiments. We ensured that the AHI variable used was consistent with AASM12 recommended rule as specified on the NCSR website. This was re-confirmed through private correspondence with the NCSR administrators. We have added an extensive section "Scoring Rules" that provides the necessary information. We especially described the procedure of "re-annotation" of NSRR for old databases.

2. This study needs to detail the implications of using data of which > 50% comprises recordings over two decades old. In contrast to what the authors describe as the similarities between the various devices, oximeter technologies have evolved.

The reviewer raises a good point. Although, oximeters are based on the same principles, some improvements have certainly been made over the past two decades. Yet, we seek a large dataset to train OxiNet and SHHS was the most suitable in that respect. Figure S2, highlights that indeed OxiNet required a large dataset to train. We however, acknowledge that the age of SHHS is an intrinsic limitation to our experimental design and have added the following sentences to the discussion-limitation section:

“Another limitation of this work is that the datasets used for the analysis are relatively old. SHHS for instance was recorded between 1995 and 1998. Improvements have been made to oximetry technology since then. We do expect that some incremental improvement can be reached provided we had access to a dataset making use of state-of-the-art oximeters and having a similar size to the SHHS. Indeed, although a relatively old dataset, the large size of SHHS was necessary to reach high model performance with OxiNet (Figure S2). However, we do not expect a change in the relative performance of OxiNet versus the benchmarked models (ODI, OBM) and thus our main conclusions.”

3. These issues tied to the change in datasets over time lead to the obvious question: what is common among these recordings that align with a diagnosis of OSA or its severity? What is the interpretation of these results that will advance the understanding of the underlying pathophysiology? The need for interpretability is concerning.

The need for interpretability is indeed important. Based on the reviewer comment we have decided to add this aspect to the manuscript. We have added a new section in the methods, named “Explainability”:

“Explainability is a critical aspect to ensure that the model is trustworthy and can be integrated into clinical practice. It enables the identification of the contributing factors and provides explanations for the predictions made by the model. Indeed, DL models are known for their black-box nature, making it difficult to understand how they arrive at their predictions. To that end, we adapted the algorithm proposed by Zeiler et al., named Feature Occlusion (FO) and originally proposed for image recognition. The algorithm has already been used in the context of time series prediction several times. The algorithm computes the importance score as the difference in output after replacing each contiguous region with a given baseline. We defined a region as a window of L_{region} seconds in the oximetry signal and performed the occlusion with a sliding window of size $L_{\text{region}}/2$, in order to have an importance score for each batch of $L_{\text{region}}/2$ seconds. The baseline to replace with was set to be the overall mean of the signal.

Furthermore, in the results, we have added the following sentences along with the following figure:

For model explainability, L_{region} was set to 120 seconds which is the order of magnitude of the duration of one to a few apnea events (severe desaturations typically last 30-45 seconds). The importance score is calculated as the difference between the predicted AHI on the original signal and the predicted AHI on the signal with the corresponding window replaced by a baseline SpO₂ value. In this study, the baseline SpO₂ value was set to the mean value of the entire recording. Figure 5 presents examples of three different recordings. Panel (a) displays an overnight signal, where OxiNet identifies clusters of desaturations. OxiNet leverages the temporal context of desaturation events within the overnight time series. This is in opposition to a rule based ODI detector that searches for desaturations as isolated events, i.e. independent of their temporal context. Panel (b) shows a signal segment with several apnea events. While OxiNet assigned relatively high scores there was no desaturation detected by the rule based desaturation detector. This reflects that the desaturation detector is too constrained while OxiNet may learn a variety of SpO₂ patterns associated with apnea and hypopnea events. Panel (c) shows a segment with no apnea or hypopnea respiratory event and, in agreement, relatively low OxiNet scores. The rule-based ODI detector, however, detected a desaturation that is not associated with a respiratory event. Overall, the explainability figures suggest that OxiNet provides added value over a simpler rule based desaturation detector. This is because the data-driven approach enables to better learn the representation of SpO₂ events during apnea and hypopnea across the high physiological variability of thousands of individuals used to train OxiNet. OxiNet also takes into account the

temporal context of events while classical rule based ODI detectors look at an event in an isolated manner.

4. Many studies have demonstrated the utility of machine learning methods for predicting OSA severity from oximeter tracings. While there is merit in establishing the generalizability of these predictions across multiple datasets, this is an incremental advance over the previous literature in the field.

We agree with the reviewer that many studies have investigated the feasibility of ML in screening for OSA (classification tasks) and we discuss some of the main contributions in the discussion section. Few works have aimed at estimating the AHI (regression task) from oximetry alone. More importantly, despite all these studies Massie et al. 2022 reviewed a total of 20 papers, representing a participant pool of 1,652 and reported an overall misdiagnosis rate of 39% (Massie, Frederik, et al. JCSM 18.3 (2022): 871-876.). It is our strong belief that this misdiagnosis rate stem from the fact that the algorithms developed perform well on some local test set but poorly when evaluated on external test set due to distribution shifts. In this new contribution we introduce a new deep learning algorithm, OxiNet, which is significantly and non-incrementally more robust than benchmark state-of-the-art approaches. By robust we mean high performing and generalizable. Indeed, the remaining gap that exist between OxiNet performance on the external test sets and the local test set is very likely due to Human inter rater variability in annotating apnea and hypopnea events. In a previous

study by Magalang UJ et. al. Sleep. 2013 Apr 1;36(4):591-6, the authors reported a human inter rater variability of ICC=0.95 for AHI estimation. The ICC reported on the external datasets was in the range 0.92-0.94 which is close from human inter rater variability.

Minor weaknesses:

1. Abstract: The authors skip over the relevance of home tests. Recognizing that the current home tests provide acceptable diagnostic performance is essential. Many components (e.g., nasal airflow) provide additional information in the possible attribution of localization of obstruction, or for sleep staging, which is not provided by oximetry.

We have modified the sentence in the abstract from:

“Existing home sleep tests have shown limited diagnosis performance.”

To:

“Existing home sleep tests may provide acceptable diagnosis performance but have shown several limitations.”

2. The comparison with non-clinical portable oximeters is irrelevant as regulatory organizations usually require these devices to have acceptance criteria before approving them for home use.

Following the reviewer’s comments, we have removed the mention of wearables that measure oximetry.

3. The authors scored the UHV dataset in-house, which is acceptable, but the potential bias should be acknowledged.

The UHV was indeed the only database that is not part of the National Sleep Research Resource (available on sleepdata.org). However, the protocol for annotating the PSG recordings also followed the AASM 2012 recommendations and scoring was performed by certified sleep technicians. We have emphasized that aspect in the database description section:

“The UHV was the only database that was not part of the National Sleep Research Resource (available on sleepdata.org). However, the protocol for annotating the UHV PSG recordings also followed the AASM 2012 recommendations and scoring was formed by certified sleep technicians.”

REVIEWERS' COMMENTS

Reviewer #2 (Remarks to the Author):

ok

Reviewer #3 (Remarks to the Author):

The authors have done an excellent job responding to the comments raised by the reviewers. Of particular note is the effort taken to incorporate explainability. The revised manuscript has no major concerns.